# Anticancer Activity of Measles–Mumps–Rubella MMR Vaccine Viruses against Glioblastoma

**DOI:** 10.3390/cancers15174304

**Published:** 2023-08-28

**Authors:** Zumama Khalid, Simona Coco, Nadir Ullah, Alessandra Pulliero, Katia Cortese, Serena Varesano, Andrea Orsi, Alberto Izzotti

**Affiliations:** 1Department of Health Sciences, University of Genova, Via Pastore 1, 16132 Genoa, Italy; zumama.khalid@edu.unige.it (Z.K.); nadir.ullah@edu.unige.it (N.U.); alessandra.pulliero@unige.it (A.P.); andrea.orsi@unige.it (A.O.); 2IRCCS Ospedale Policlinico San Martino, Largo Rosanna Benzi 10, 16132 Genoa, Italy; simona.coco@hsanmartino.it (S.C.); serena.varesano@hsanmartino.it (S.V.); 3Department of Experimental Medicine, University of Genoa, 16132 Genoa, Italy; cortesek@unige.it

**Keywords:** oncolytic viruses, virotherapy, glioblastoma, MMR vaccine

## Abstract

**Simple Summary:**

This research has been suggested after a gap was found in the previous literature, and our hypothesis of the MMR vaccine being an oncolytic virus. The present study has been designed to study and cover the followings objectives: to evaluate the therapeutic effect of the measles virus vaccine strains on cancer, through wet-lab experimental analysis; and to study the previous literature on the application of oncolytic viruses. Our findings highlight the therapeutic potential of the MMR vaccine strain for the treatment of glioblastoma (GBM).

**Abstract:**

Background: Oncolytic viruses (OVs) have been utilized since 1990s for targeted cancer treatment. Our study examined the Measles–Mumps–Rubella (MMR) vaccine’s cancer-killing potency against Glioblastoma (GBM), a therapy-resistant, aggressive cancer type. Methodology: We used GBM cell lines, primary GBM cells, and normal mice microglial cells, to assess the MMR vaccine’s efficacy through cell viability, cell cycle analysis, intracellular viral load via RT-PCR, and Transmission Electron Microscopy (TEM). Results: After 72 h of MMR treatment, GBM cell lines and primary GBM cells exhibited significant viability reduction compared to untreated cells. Conversely, normal microglial cells showed only minor changes in viability and morphology. Intracellular viral load tests indicated GBM cells’ increased sensitivity to MMR viruses compared to normal cells. The cell cycle study also revealed measles and mumps viruses’ crucial role in cytopathic effects, with the rubella virus causing cell cycle arrest. Conclusion: Herein the reported results demonstrate the anti-cancer activity of the MMR vaccine against GBM cells. Accordingly, the MMR vaccine warrants further study as a potential new tool for GBM therapy and relapse prevention. Therapeutic potential of the MMR vaccine has been found to be promising in earlier studies as well.

## 1. Introduction

An oncolytic virus is a naturally occurring or genetically engineered virus that may specifically reproduce in cancer cells and destroy them without harming healthy cells [1]. In contrast to gene therapy, oncolytic virotherapy uses the virus as an active pharmaceutical agent rather than only a carrier, as in transgene delivery. Between 1950 and 1980, several clinical trials were carried out employing wild-type or naturally attenuated viruses to treat cancer, including hepatitis, adenoviruses, West Nile disease, yellow fever, and dengue fever viruses [2]. However, these viruses were not considered therapeutic agents because there was no established technique to decrease virulence while maintaining viral replication in cancer cells at that time. Oncolytic virotherapy may be more effective when combined with immunotherapy or chemotherapy. It may potentially be useful to equip oncolytic viruses with immunostimulatory genes or cancer therapy genes. As far as the mechanism of action of oncolytic virus is concerned, it takes over the tumor cell’s protein factory after infection and prevents tumor cells from making enough protein to meet their nutritional requirements, effectively disrupting the tumor cell’s normal physiological process [3,4,5] (Figure 1). Against different cancers, various DNA and RNA-containing viruses have been studied which are considered oncolytic viruses. They include adenovirus, Herpes Simplex Virus (HSV), rubella virus, measles virus, vaccinia, coxsackievirus, reovirus, etc. [6]. Other oncolytic viruses for cancer therapy include rhabdovirus, Seneca Valley Virus, parvovirus, retroviruses, etc. [7]. The German measles (Rubella) seems to be a potential candidate for innovative cancer treatment because of its use in vaccination in combination with measles and mumps [8]. However, extensive studies are needed to be performed to conclude this as a scientific fact. Different viruses have shown good, exceptional anticancer efficacy against various cancer types. Oncolytic viruses use various mechanisms to destroy cancer cells and enhance their therapeutic effects [9]. Several oncolytic viruses have been used in the treatment of GBM, including several modified adenoviruses, specifically. Intraarterial delivery of oncolytic viruses has been performed in the hepatic circulation and is now being studied in the cerebral circulation to help increase delivery [10]. As shown in Table 1, many RNA viruses have been utilized in oncolytic treatment clinical studies to treat various forms of cancer. The personalized utilization of oncolytic viruses against GBM may improve the response rates, based on specific tumor- or patient-related characteristics [11]. Our research focused primarily on the MMR vaccine in cancer virotherapy employing glioblastoma cancer, which is thought to be extremely aggressive and resistant to apoptosis using standard anticancer treatments [12]. Recent evidence has pointed out that tumor-associated macrophage/microglia involvement is an important factor contributing to oncolytic viruses’ treatment failure [13]. Several oncolytic viruses have been developed and validated in clinical trials with favorable safety profiles and efficacy against GBM. Recently, the Zika virus was shown to preferentially target and kill GBM stem cells, promising therapeutic effects in preclinical models [14]. However, newly emerging evidence of the cancer stem cell plasticity challenges this hypothesis by proposing that the cancer stem cell pool can be regenerated from non-cancer stem cells post-treatment [15]. Synthetic biology is an emerging field that focuses on the development of synthetic DNA constructs that encode networks of genes and proteins to perform new functions [16].

## 2. Materials and Methods

### 2.1. Cell Lines and Cell Culture

Human glioblastoma (GBM) cell lines (U87MG, U138MG) provided by Cell Bank, Ospedale San Martino, Genova, Italy, were used. Both cell lines were cultured in Eagle’s Minimum Essential Medium (MEM with 1 mM Sodium Pyruvate) supplemented with 10% of fetal bovine serum (FBS), 2 mM L-glutamine, and 100 IU/mL penicillin-streptomycin (Euroclone, S.p.A., Milan, Italy). These cell lines were maintained at 37 °C in a humidified atmosphere containing 5% CO_2_. Another two types of primary GBM cells (GBM10, and GBM23) were derived from patients (Ospedale San Martino, Genova, Italy). Both patients had their first surgery and had never received chemo or radiotherapy [58]. The glioma post-surgical specimens were obtained from the Neurosurgery Department of IRCCS Ospedale San Martino (Genova, Italy) after patients’ informed consent and Institutional Ethical Committee approval (CER Liguria register number 360/2019).

Cultures were maintained in flasks coated with Matrigel™ (1:200; BD Biosciences, Erembodegem, Belgium) in 50% Neurobasal medium, 50% DMEM/F12 medium, 1X B27 Supplement (Life Technologies, Ghent, Belgium), 10 ng/mL bFGF and 20 ng/mL EGF (PeproTech Inc., Westlake Village, CA, USA), 2 mM glutamine (Life Technologies, Ghent, Belgium), and 2 µg/mL heparin (Sigma Aldrich, St. Louis, MO, USA). A normal mice microglial cell line (BV2) was used as a control. This cell line was semi-adherent and cultured in RPMI 1640 medium with 10% FBS, 2 mM L-glutamine, and 100 IU/mL penicillin-streptomycin (Euroclone, S.p.A., Milan, Italy) and maintained at 37 °C in a humidified atmosphere containing 5% CO_2_. The morphology of all the cultures was observed using a Zeiss Axiovert 10 microscope and a Nikon Digital sight DS-5M camera.

### 2.2. Vaccine Treatment

To check the morphology of cells with and without treating them with the MMR vaccine, commercially available vaccine; M-M-R VaxPro (EMEA/HC/000604, Italy) was used. Cells were cultured in T25 flasks and after achieving 70% confluency, they were treated with the vaccine.

### 2.3. MTT Assay

To evaluate the viability of cells with and without the treatment of the MMR vaccine, cells were grown with a seeding density of 0.01 × 10^6^ in a 96-well plate. At 70% cell confluency, they were infected with the vaccine in triplicates. Then, at 72 h post-infection, 11 μL of MTT (3-(4,5-dimethylthiazol-2-yl)-2,5-diphenyltetrazolium bromide, Sigma-Aldrich, St. Louis, MO, USA) in 2 mg/mL-PBS (Phosphate Buffer Saline) (Euroclone, S.p.A., Milan, Italy) was added to each well. After 4 h incubation at 37 °C, the precipitates were dissolved in 111 μL of 10% SDS (0.01 M HCL) overnight. The plates were then analyzed on an ELISA reader at 570 nm. Absorbance recorded in uninfected cells was assumed to represent 100% cell viability. 

### 2.4. Transmission Electron Microscopy (TEM)

Due to having the least viability among other GBM cell samples, U87MG cells were processed for TEM. Cells were grown to 70% confluency in a T25 flask and infected with the MMR vaccine. After 72 h, the cells were fixed in 4% PFA (Paraformaldehyde) (Sigma-Aldrich, St. Louis, MO, USA) in PBS (Euroclone, S.p.A., Milan, Italy) to be processed for TEM.

### 2.5. Cell Cycle Study by Flow Cytometry 

All the cells were cultured in 24-well plates with a seeding density of 0.05 × 10^6^ in each well. After 24 h with 60–70% confluency, they were infected with the MMR vaccine. After 72 h post-infection, cell samples were prepared for cell cycle assay using the kit protocol of the Muse Cell Cycle Kit (Cat No. MCH100106). Samples were trypsinized and centrifuged at 1500 rpm for 5 min. Pellets were washed with PBS. Then, pellets were resuspended in chilled 70% ethanol drop-by-drop while they were vortexed. These fixed pellets were then stored at −20° until the time of assay. For assay, cells were centrifuged at 4°, 300× *g* for 5 min, and ethanol was removed. Pellets were washed and then resuspended in 200 μL of PBS and then centrifuged for another 5 min at 300× *g* at 4°. Then, PBS was removed, and pellets were resuspended in 200 μL of muse cell cycle reagent and incubated at room temperature in darkness for 30 min. After incubation, cells were analyzed using the Muse Cell Cycle Analyzer to check in which phase of cell cycle they were arrested. Values were expressed as the percentage of cells in the G0/G1, S, and G2/M phase of cell cycle.

### 2.6. Viral Load Test

The extraction of the cell samples was performed using the MagCore^®^ Viral Nucleic Acid Extraction (RBC Bioscience Corp. Taiwan) following the manufacturer’s instructions (Cartridge Code: 202; elution volume: 60 µL) by means of MagCore^®^ HF16 Plus automated nucleic acid extractor (RBC Bioscience Corp. Taiwan). In total, 400 µL of the sample was transferred to a sample tube adding 20 µL of proteinase K, 10 µL of the carrier, and 10 µL of the respective internal controls (IC) specific for the subsequent analysis of measles, mumps, and rubella. The CE-marked RealCycler SARA-UX kit (Valencia, Spain) and RealCycler MuV kit of Progenie Molecular (Valencia, Spain) were used for the molecular analysis of measles and mumps, respectively. Then, 8 µL of eluates were added to the ready-to-use mix before starting the RT-PCR. We performed the PCR protocol following the manufacturer’s instructions. The CE marked Sacace™ Rubella Real-TM Qual was used for the molecular analysis of rubella. Then, 10 μL of the eluate was added to the appropriate tubes with the reaction mix before starting the RT-PCR. We performed the PCR protocols on a CFX96™ thermal cycler (BioRad, Hercules, CA, USA) following the manufacturer’s instructions. The results were interpreted by the software of the PCR instrument. The results of the analysis were considered reliable only if the results obtained for positive and negative controls of amplification were correct. The threshold line was set automatically for all, using manufacturer’s instructions.

## 3. Results

### 3.1. Morphological Analysis

Glioblastoma (GBM) cell lines (U87MG, U138MG), primary GBM cells (GBM23, GBM10), and normal microglial cells (BV2) after 72 h following the MMR vaccine treatment were observed under an inverted microscope. Cell morphology was present both in control (C) and MMR-treated (T) cells. U87MG-untreated cells aggregate in an organoid-like morphology while U138MG cells grew independent to each other (Figure 2a, left panels).

A significant change was observed after 72 h of infection in the glioblastoma cell samples. U87MG-T(MMR-treated cells) were seen unattached from the flask surface, aggregated, and disturbed with many dead cells in comparison with the control. In U138MG cells, morphology was not grossly disturbed but many dead cells were found in the MMR-treated cells (Figure 2a, lower panels). The differences among MMR-treated and control-untreated cells from primary glioblastoma (GBM23, GBM10) were less evident than in cell lines. Indeed, only a small increase in the number of cells unattached from the flask surface was detected (Figure 2b). In normal microglial cells (BV2) no specific change was observed in morphology after 72 h of MMR infection (Figure 2c).

### 3.2. Viability Analysis

After performing MTT assay for viability, absorbance was recorded for all cell samples in the ELISA reader. A significant decrease in viability was observed in MMR-infected glioblastoma cells (T) as compared to the assumed 100% viability of control cells (C); however, in normal microglial cells, there was not any significant viability difference between control (C) and treated (T) cells.

At 72 h since MMR treatment, in normal microglial cells (BV2), 87% of the cells remained viable (1.1-fold decrease only) after 72 h post-infection with MMR vaccine, hence proving the MMR vaccine safe to normal brain cells (Figure 3a). The highest viability decrease was observed in U87MG MMR treated with a viability of 11 ±< 0.01%, and of GBM10 with a viability of 12.5 ±< 0.01%. Other cells’ viability were U138MG- 29 ±< 0.01%, and GBM23 44 ±< 0.01%. Accordingly, 72 h after MMR treatment, cell viability was decreased by 9.1-fold in U87MG, 3.4-fold in U138MG, 8.0-fold in GBM10, and 2.3-fold in GBM23 (Figure 3b,c). These differences were statistically significant as evaluated by the nonparametric Kruskal–Wallis test (*p* < 0.05). These results indicate that the MMR vaccine can kill glioblastoma cells (Figure 3).

### 3.3. Transmission Electron Microscopy (TEM)

To further assess the morphology of the least viable cells (U87MG) after infection with MMR vaccine, Transmission Electron Microscopy was comparatively performed in infected and control cells 72 h after MMR treatment. Infected cells showed increased volumes of whole-cell, nucleus, and rough endoplasmic reticulum as compared to control U87MG cells (Figure 4a,b). Further signs of cell degeneration detected were cell membrane disruption and disturbed structure of organelles mainly referring to mitochondria. Multiple copies of the virus (both whole virions and components) were detected inside the infected cells (Figure 4c). These findings suggest that the MMR vaccine has taken charge of the cancer cell and used its replication, metabolic, and protein synthesis machineries to replicate inside the cells. This situation resulted in cancer cell sufferance and necrosis. 

### 3.4. Cell Cycle Study by Flow Cytometry

To assess DNA content at different cell stages, MUSE cell cycle assay was performed. Assay was performed using control (untreated) and MMR-infected cells (cell lines, primary cells, and normal microglial cells) to check at which stage of cell cycle they were blocked after 72 h of infection. Obtained results are reported in Table 2.

After MMR vaccine treatment, glioblastoma cells underwent cell cycle blockage as demonstrated by the increased number of cells in G0/G1 and the decreased number of cells in G2/M. The most sensitive cells to these effects were GBM10-T and U138G-T. A similar effect was also observed in normal microglial BV2-T cells (Figure 5). 

### 3.5. Viral Load Test

Analysis of intracellular viral loads in each cell samples was performed by PCR to evaluate cell sensitivity to intracellular viral penetration. RNA analysis of the measles virus indicates a sensitivity of cell samples to viral penetration in this decreasing order: GBM23, U138MG, U87MG, and GBM10. For the mumps virus, the sensitivity of cell lines to viral penetration in decreasing order was: U87MG, U138MG, GBM10, and GBM23. In the rubella virus, the sensitivity of cell lines to viral penetration in decreasing order was: U138MG, GBM10, U87MG, and GBM23 (Figure 6). The positivity of amplification curves was detected only in MMR-treated samples, while negative results were obtained in control (untreated) samples. In normal microglial cells (BV2), the intracellular viral load was lower than that detected in GBM cells as demonstrated by the PCR higher CT threshold (Figure 7, Figure 8 and Figure 9). Quantitative data dealing intracellular viral load in each cell sample were plotted based on the CT value (PCR cycle positivity threshold).

These results indicated that glioblastoma cancer cells are highly sensitive to infection by mumps and measles viruses and, only to a lesser extent, to rubella virus. Accordingly, the main contribution to cell cytopathic effects detected was provided by mumps and measles viruses while the rubella virus mainly contributed to cell cycle arrest (Figure 7). Normal microglial cells (BV2) were less sensitive than GBM cells to MMR infection as inferred from comparisons of PCR Ct thresholds (Figure 9). Indeed, the lowest Ct were (a) for measles virus it was 19 in GBM cells and 33 in normal BV2 cells; (b) for mumps virus it was 20 in GBM cells and 27 in normal BV2 cells; and (c) for rubella virus it was 27 in GBM cells and 33 in normal BV2 cells (Figure 8 and Figure 9). 

## 4. Discussion

The oncolytic activity of measles, mumps, and rubella vaccine viruses detected in our study is specifically exerted only towards cancer cells. Indeed, it is well established that these vaccine viruses do not have any possibility of inducing adverse effects in normal cells. In our experiments using normal microglial cells (BV2), no obvious reaction was observed in MMR-treated BV2 cells. Msaouel et al. also reported clinical trials with oncolytic measles virus (MV-CEA) in mice, using normal brain cells as a safety control [59]. This study employed the transgenic mouse model Ifnarko CD46Ge to assess the neurotoxicity of intracranial injection of MV-CEA to a normal brain. These animals express CD46 receptors, lack interferon receptors, and are vulnerable to MV replication. Notably, the FDA has also acknowledged IFNARko-CD46Ge mice as animal toxicology models to study oncolytic measles viruses [60]. The MV entry receptor CD46 has an expression like humans, the knockout of the interferon receptor facilitates the replication of measles viruses, which are otherwise strongly restricted by type I interferon. The stereotactic parameters for the orthotopic efficacy trial were used to give MV-CEA at the same dosage, volume, and timing. After giving the certain dose of MV-CEA, treated mice were monitored for up to three weeks in comparison with sham-treated (saline) and untreated mice as controls. MV-CEA intracerebral administration to transgenic mice did not cause any neurotoxicity. No neurological, clinical, or behavior harm was observed at any stage during the investigation [59]. In a recent report published in *Nature Medicine*, a genetically engineered virus was injected directly into a patient’s brain tumor. An overall 12-month survival of 52.7% was recorded, which is significantly higher than the 20% prespecified efficacy threshold. No cell toxicity was observed during this experiment [61].

Additionally, in safety trials of MMR vaccine in humans, rare serious outcomes have been observed in adults [62]. Regarding the safety of the MMR vaccine used in our study, European Medicine Agency has conducted various safety trials of this vaccine in adults and children no younger than 9 months and found no toxicity [63].

The most severe brain tumor still characterized by a very poor prognosis is glioblastoma (GBM), which is one of the most deadly cancers. The common survival time is only one year since diagnosis, and only approximately 5% of patients survive for five years. The current accepted standard of care for GBM is maximum safe surgical resection followed by adjuvant concurrent chemo-radiation therapy and adjuvant chemotherapy with temozolomide. However, rather than obtaining full recovery from GBM, this treatment only extends patients’ lives [64]. Accordingly, patients with GBM still have a very dismal prognosis after multimodal therapy. Because of this situation, there is an urgent need to develop innovative therapies for GBM. Clinical trials using oncolytic viruses have demonstrated significant results, albeit in a limited proportion of GBM patients, and are being considered as a new treatment for this patient population [11]. A study performed by Appolloni and colleagues used R-613 oncolytic HSV (herpes simplex viruses) to check its efficacy in glioblastoma treatment and found this a promising approach [65]. In another study by Reisoli and colleagues, HER2 retargeted HSVs were used as a therapeutic approach using mice models against high-grade glioma and observed significant results [66].

In the current study, the MMR (Measles, Mumps, and Rubella) vaccine was used as an oncolytic virus treatment on glioblastoma cells. Control and vaccine-treated glioblastoma cell samples along with normal microglial cells were comparatively examined for morphology, viability, cell DNA content at various cell-cycle phases, and viral RNA intracellular load. Obtained results supported our hypothesis that the MMR vaccine has an oncolytic activity towards GBM but is safe for normal microglial cells. After the exposition of cell lines to the MMR vaccine for 72 h, the morphology of glioblastoma (GBM) cell lines (U87MG, U138MG) and primary GBM cells (GBM23, GBM10) underwent considerable alterations, especially in the U87MG cell line. All four types of GBM cells were disturbed, aggregated, and detached from the flask in comparison to the control group. Conversely, normal microglial cells (BV2) retained almost the same morphology in untreated and MMR-treated cell samples after 72 h. The study reported by Allen and colleagues [67] on the treatment of gliomas, also suggested that the use of oncolytic measles virus strains is a unique and effective anticancer approach. This virus is now undergoing Phase I testing in recurrent glioblastoma patients after demonstrating preclinical effectiveness following orthotopic treatment in numerous glioma models and safety tests in a transgenic mouse model and primates [60]. In a recent study, glioblastoma was exposed to the adeno virus, and the results of in vitro evaluation of Ad6’s oncolytic abilities against U87 and U251 GB cell lines were equivalent to those of Ad5. Both Ad5 and Ad6 (adenoviruses) exhibited a cytotoxic impact on U87 (*p* = 0.01 and *p* = 0.05, respectively) and U251 (*p* = 0.01) cell lines when treated with a dosage of 1 lg (TCID50/cell) [68]. In comparison to the presumed viability of 100% in control cells, a considerable shift in viability was observed in infected cancer cell samples in our results. The treated U87MG cell line showed more MMR vaccine cancer-killing efficacy than other cell lines, with a viability of 11%, compared to: 12.5% for GBM10 cells, 29% for U138MG cell line, and 44% for GBM23 cells. However, normal microglial cells (BV2) did not show an obvious shift in viability after 72 h of MMR vaccine treatment. In total, 87% of BV2 cells were found viable after 72 h of treatment, 13% cells might have died due to nutrient deficiency. These findings suggest that the MMR vaccine has a considerable chance of eliminating glioblastoma cells but is safe for normal brain cells. The analyses of viral load of each virus of MMR vaccine in all cell line types showed that: (a) maximum measles load was measured in GBM-23 cell line; (b) maximum mumps load was observed in U87MG cell line; and (c) maximum rubella load was found in the U138MG cell line. All glioblastoma cell samples were found highly sensitive to infection by mumps and measles viruses and, to a lesser extent, to rubella virus. Normal microglial cells (BV2) were less sensitive to MMR infection in comparison with the GBM cells. The main contribution to cell cytopathic effects was provided by the mumps and measles viruses, while rubella viruses mainly contributed to cell cycle arrest. The effects of rubella virus in various cell lines have been studied in past. It has been found highly genotoxic during the first trimester of the gestational period [69]. In combination with graphene oxide to check viral infection in adenocarcinoma, cytotoxicity of rubella virus has been observed by Kuropka and colleagues [70]. However, our findings indicate that there is a variability in GBM susceptibility to MMR viruses’ infection. Accordingly, a mixture of all three MMR viruses should be used to have an efficacy arrest of GBM cell growth. The use of only one viral strain (as insofar performed) likely underestimates the chances offered by MMR viruses in arresting GBM cell growth. The ability of MMR viruses to infect, kill, and arrest GBM cancer cells is likely since these viruses are also highly effective in infecting human fetal cells. Indeed, the ability of wild (not vaccine) MMR viruses to induce fetal sufferance in infected mothers is well established with reference to glial cells. Indeed, defects of the central nervous system such as encephalitis and ipo and anencephalia are well known consequences of wild MMR virus infection [71]. Fetal cells share many antigens with cancer cells, as detailed in Table 3. 

## 5. Conclusions

In conclusion, our study showed that the MMR vaccine exerts cancer-killing activity by decreasing the viability of and modifying the morphology and growth of GBM cancer cells. Multiple copies of viruses were found inside the cell membrane and nucleus of the treated GBM cells by Transmission Electron Microscope and PCR viral load analyses. Measles and mumps viruses induced direct cytopathic effect while rubella virus induced arrest of cell replication. Indeed, DNA replication of cancer cells decreased significantly after treating with MMR. Accordingly, the MMR vaccine warrants further study as a potential new tool for GBM therapy and relapse prevention. This strategy is of particular interest since MMR vaccine viruses have been already tested for their safety for a long time, on a huge number of subjects also including fragile subjects such as children. Additionally, in our study, the MMR vaccine did not affect normal microglial cells in terms of morphology, viability, and intracellular viral load. The MMR vaccine could represent a new effective and safe tool for GBM therapy characterized by a remarkable compliance to be used in addition to standard chemo-radiotherapy. 

## Figures and Tables

**Figure 1 cancers-15-04304-f001:**
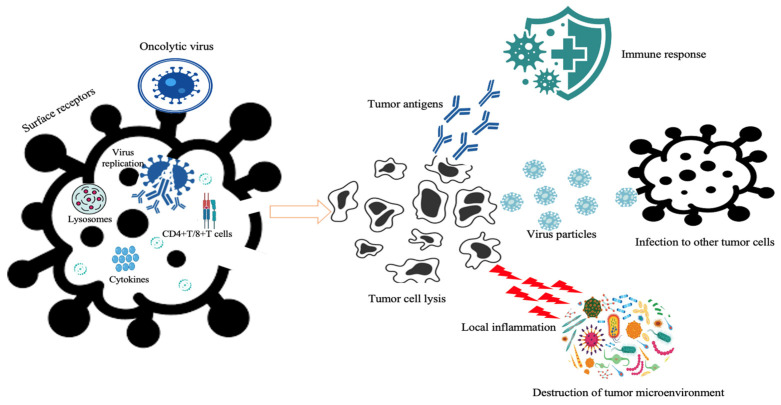
Mechanism of action of oncolytic viruses.

**Figure 2 cancers-15-04304-f002:**
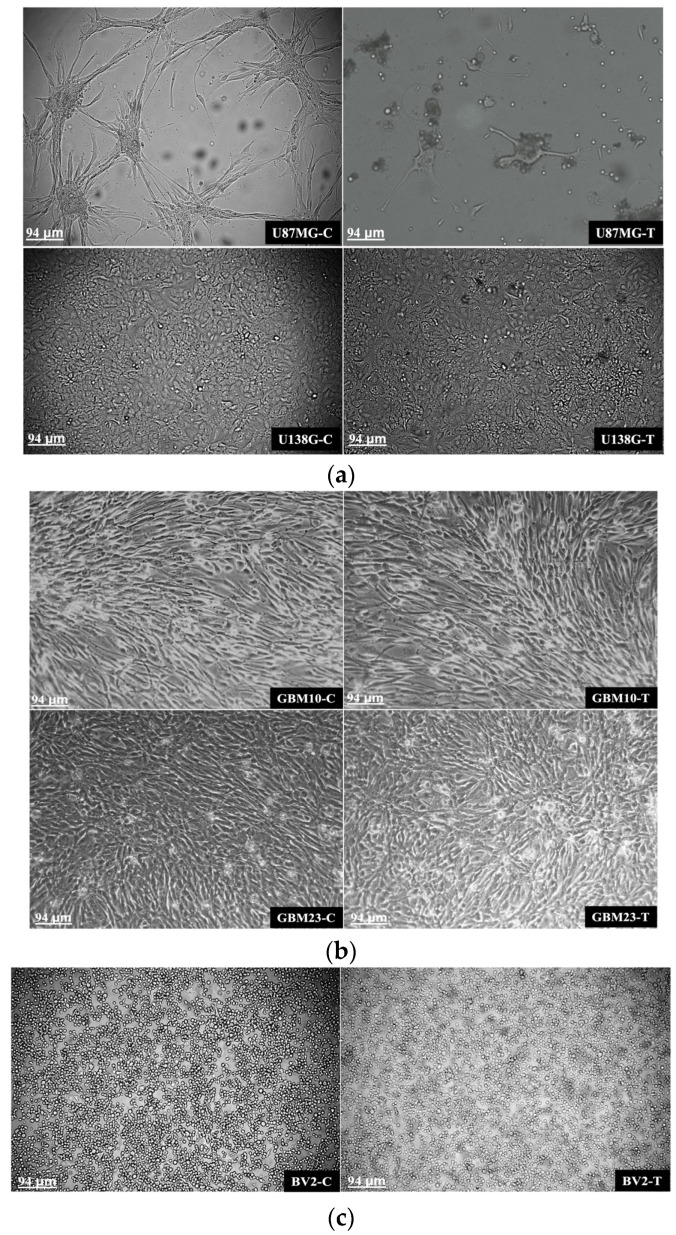
Microscopic images of control (C) and MMR vaccine-treated glioblastoma cells (T) after 72 h since MMR vaccine infection. (**a**) Glioblastoma cell lines; treated U87MG with unattached cells and grossly disturbed morphology; less evident alterations were observed in U138MG-treated cells (upper right panel) (size bar 94µm). (**b**) Primary glioblastoma cells neither control (C) nor MMR- treated (T) showed remarkable alteration after MMR vaccine treatment but there was only a small increase in the number of unattached cells (size bar 94µm). (**c**) Normal mice microglial cells neither control (C) nor MMR-treated (T) showed significant alteration in morphology after 72 h (size bar 94 µm).

**Figure 3 cancers-15-04304-f003:**
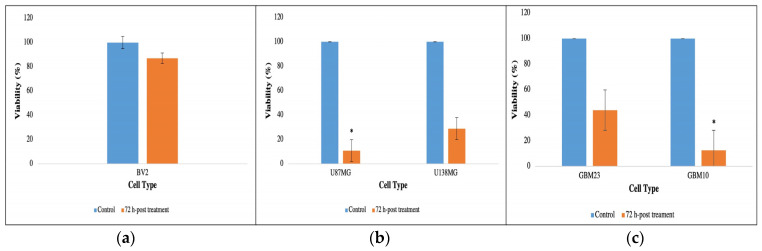
Glioblastoma cell viability (vertical axis) in glioblastoma cells either untreated (Control, blue columns) or treated with MMR vaccine for 72 h (orange columns). (**a**) Cell viability of normal mice microglial cells (BV2) before and 72 h after MMR vaccine treatment. (**b**) Cell viability of GBM cell lines (U87MG and U138MG) before and 72 h after MMR vaccine treatment. (**c**) Cell viability of primary GBM cells (GBM23, GBM10) before and 72 h after MMR vaccine treatment. A remarkable decrease in cell viability was observed in all glioblastoma cells post-MMR vaccine treatment as compared to normal microglial cells. (*: *p* < 0.05).

**Figure 4 cancers-15-04304-f004:**
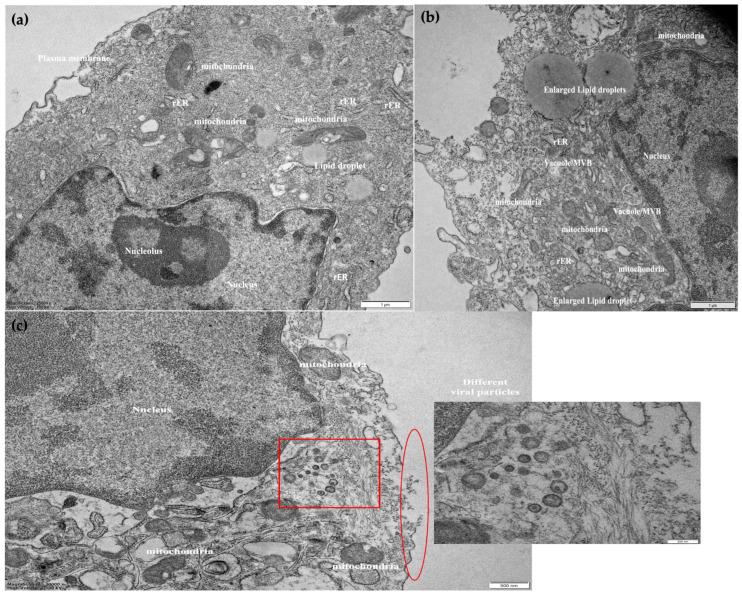
Transmission Electron Microscopy (TEM) images of U87MG (Magnification-30,000×) (**a**) untreated U87MG; (**b**) U87MG 72 h after MMR infection; (**c**) magnified image of disrupted cell membrane (red circle) and multiple copies of virus particles inside the cell (red square).

**Figure 5 cancers-15-04304-f005:**
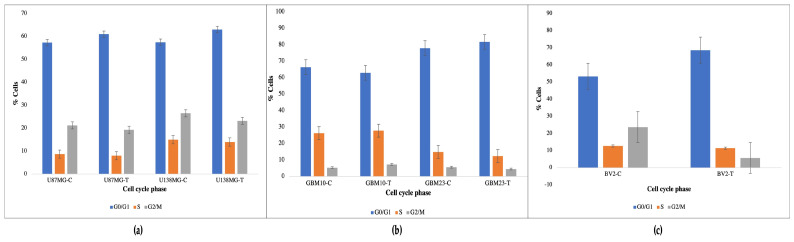
Effect of MMR vaccine (Control-C, MMR Vaccine Treated-T) on cell cycle in (**a**) cell lines, (**b**) primary cell cultures, and (**c**) normal microglial cells. Vertical axes indicate the percentage of cell at each cell cycle stage (G0/G1 blue, S orange, and G2/M grey).

**Figure 6 cancers-15-04304-f006:**
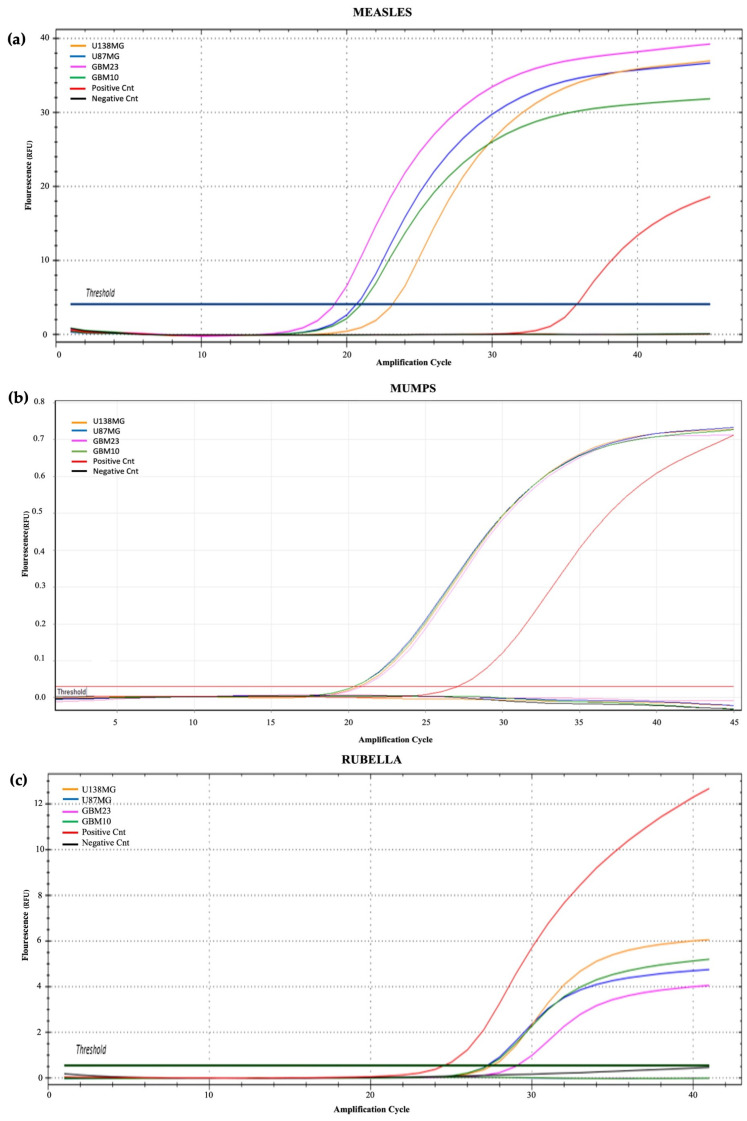
Amplification curves of intracellular loads of MMR viruses. (**a**) measles, (**b**) mumps, and (**c**) rubella. Cells tested were U138MG (orange), U87MG (blue), GBM23 (pink), and GBM10 (green), the positive (red) and negative (black) control. Results indicate that glioblastoma cell samples are more sensitive to infection by measles and mumps viruses and, to lesser extent, to rubella virus.

**Figure 7 cancers-15-04304-f007:**
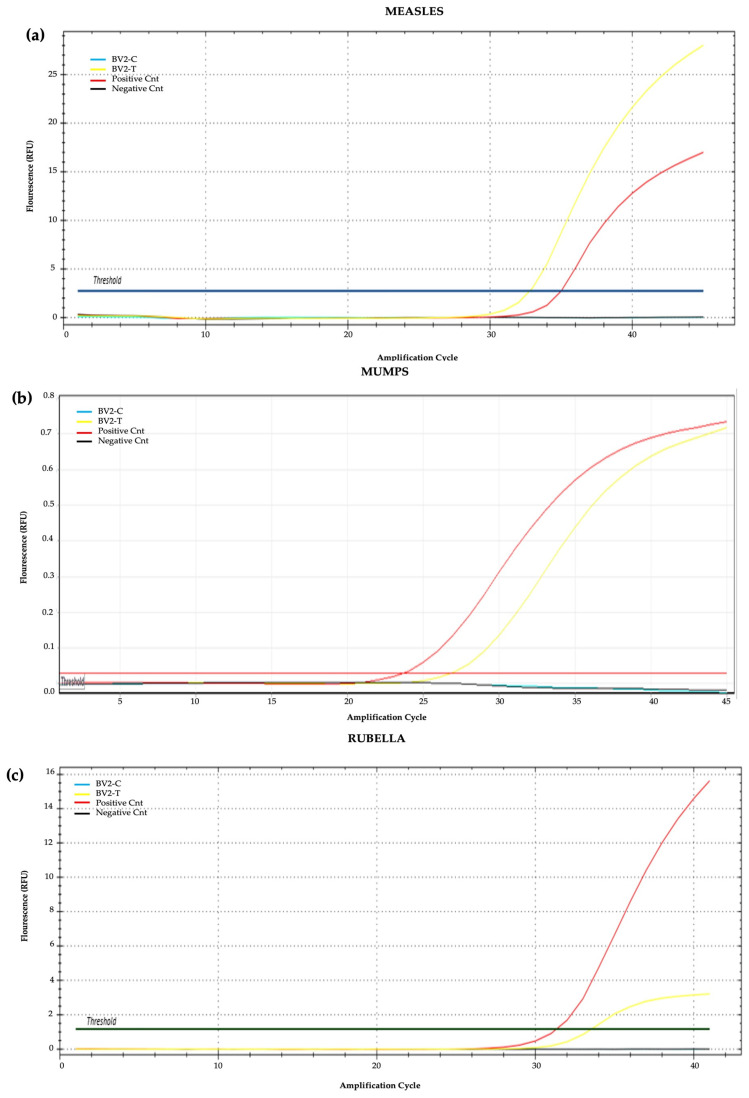
Amplification curves of intracellular loads of MMR viruses in normal microglial cells. (**a**) measles, (**b**) mumps, and (**c**) rubella. Samples tested were BV2-C (blue), BV2-T (yellow), positive (red), and negative (black) controls.

**Figure 8 cancers-15-04304-f008:**
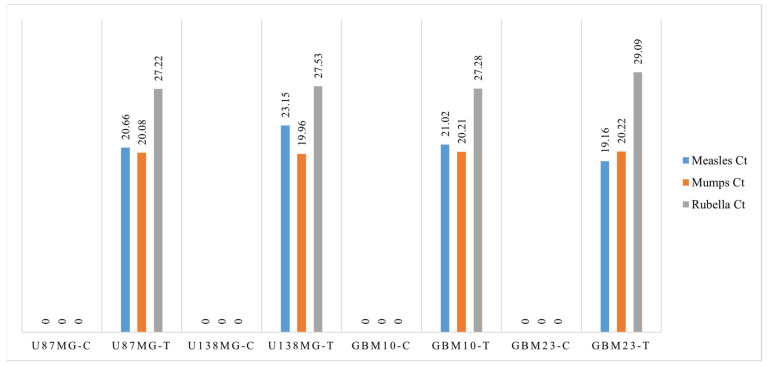
The graph reports the columns of Ct values obtained by PCR molecular analysis of measles (light blue), mumps (orange), and rubella (grey) viruses penetrated inside U87MG, U138MG, GBM10, and GBM23 cell samples. Negative results were obtained in untreated control samples (C).

**Figure 9 cancers-15-04304-f009:**
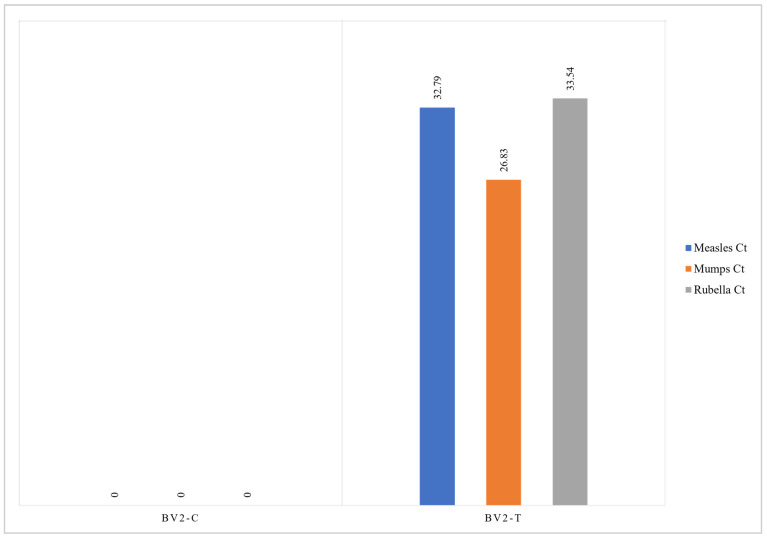
The graph represents the Ct values obtained by PCR analyzing measles (light blue), mumps (orange), and rubella (grey) viruses penetrated inside normal microglial cells (BV2-T) after MMR vaccine infection. Negative results were obtained in untreated control BV2 cells (BV2-C).

**Table 1 cancers-15-04304-t001:** Oncolytic RNA viruses used in phase I, II, and III clinical trials.

Phase of Trial	Type of Cancer	Virus	Comments	Reference
Phase I Intradermal injection	Melanoma	ND (Newcastle disease)	The use of an autologous ASI vaccination as an adjuvant treatment for melanoma patients did not show clinical efficacy in this study.	[17]
Phase I, intravenous administration of PV701	Advanced solid cancers (A malignant solid tumor that has progressed to other parts of the body or is no longer responding to therapy)	PV701 ND	PV701 should be investigated further as a potential new cancer treatment. Over the course of 195 cycles, a 100-fold dose intensification was obtained. For outpatient dosing, a first dose MTD of 12 109 plaque-forming units (PFU)/m^2^ was established. Patients tolerated an MTD for consecutive doses of 120 109 PFU/m^2^ after an initial dose of 12 109 PFU/m^2^.	[18]
Phase I optimized clinical regimen for the oncolytic virus PV701	Advanced cancers	PV701	Patient tolerance was enhanced with slow infusion, and the first dose was safely increased compared to two earlier PV701 trials. This slow infusion regimen was chosen for further PV701 clinical research due to enhanced tolerability and encouraging indicators of efficacy.	[19]
Phase I trial of cyclophosphamide as an immune modulator for optimizing oncolytic reovirus delivery to solid tumor	Advanced solid tumors	RT3D (Reovirus Type 3 Dearing)	Cyclophosphamide coadministration with reovirus is safe, however it does not reduce host antiviral responses. Alternative immunomodulation techniques should be investigated, however reovirus’s interaction with PBMCs may allow it to survive and elude neutralizing antibodies even at high levels.	[20]
Phase I study of the combination of intravenous reovirus type 3 Dearing and gemcitabine in patients with advanced cancer	Advanced solid cancers	RT3D	Reovirus can be safely coupled with full dose gemcitabine at a level of 1 × 1010 TCID50. The combination of reovirus with gemcitabine alters the neutralizing antibody response, which may have an impact on the treatment’s safety and efficacy.	[21]
Phase I study of intravenous oncolytic reovirus type 3 dearing in patients with advanced cancer	Advanced cancers	RT3D (Reovirus Type 3 Dearing)	Oncolytic reovirus can be safely and routinely given as an i.v. injection at dosages up to 3 × 10^10^ TCID50 for 5 days every 4 weeks without causing severe side effects. ReovirSal infection of metastatic tumor deposits was found to be effective. Reovirus is a safe agent that should be studied further in phase II trials.	[22]
Phase I trial of intertumoral administration of reovirus in patients with histologically confirmed recurrent malignant gliomas	Recurrent malignant gliomas	Reolysin (Reovirus)	In patients with recurrent malignant gliomas (MGs), intratumoral injection of the genetically unaltered reovirus was well tolerated and there was no grade III or IV adverse events (AEs) that could have been caused by the treatment.	[23]
Phase I trial of percutaneous intralesional administration of reovirus type 3 dearing (Reolysin^®^) in patients with advanced solid tumor	Advanced solid tumors	Reolysin (Reovirus)	This unattenuated oncolytic reovirus’ good safety profile, lack of viral shedding, and potential therapeutic action have made it an appealing cancer therapeutic agent for ongoing clinical research, notably in the setting of locally progressed accessible cancer for symptom palliation.	[24]
Phase I trial of single agent reolysin in patients with relapsed multiple myeloma	Multiple myeloma	Reolysin (Reovirus)	In the use of a single-agent treatment within multiple myeloma cells, reolysin was well tolerated and linked with ardent reoviral RNA myeloma cell entrance but only minor intracellular reoviral protein synthesis. There findings suggest that, like other malignancies, Reolysin-induced oncolysis in multiple myeloma cells necessitates combined therapy.	[25]
Phase 1 clinical trial of intertumoral reovirus infusion for the treatment of recurrent malignant gliomas in adults	Malignant glioma	Reolysin (Reovirus)	There was no evidence of dose-limiting toxicity, and no maximum tolerable dose was reached. Some patients showed signs of antiglioma action. This first report of reovirus intratumoral infusion in patients with recurrent malignant glioma found the procedure to be safe and well tolerated, indicating that more research is needed.	[26]
Phase I trial and viral clearance study of reovirus (Reolysin) in children with relapsed or refractory extra-cranial solid tumors	Extracranial solid tumors	Reolysin (Reovirus)	Reolysin was well tolerated in children when given alone or in combination with oral cyclophosphamide at a dose of 5 × 108 TCID50/kg daily for 5 days. The virus was quickly eliminated from the serum, and there was no evidence of shedding in the stool or saliva.	[27]
Recurrent glioblastoma treated with recombinant poliovirus	Glioblastoma	PVSRIPO (Polio Virus)	PVSRIPO infusions into the tumor confirmed the absence of neurovirulent potential in patients with recurrent WHO grade IV malignant glioma. At 24 and 36 months, the survival rate of patients who received PVSRIPO immunotherapy was higher than that of historical controls.	[28]
Immunological effects of low-dose cyclophosphamide in cancer patients treated with oncolytic adenovirus	Advanced solid tumors resistant to chemotherapy	Ad5/3-(delta)24 (adenovirus)	They conclude that low-dose CP has immunological effects that make it a good candidate for oncolytic virotherapy. While the results of this first-in-human study imply good safety, intriguing efficacy, and long survival, they should be validated in a randomized trial.	[29]
Phase I clinical trial of Ad5/3-∆24, a novel serotype-chimeric, infectivity-enhanced, conditionally replicative adenovirus (CRAd)	Ovarian Cancer	Ad5/3-(delta)24 (adenovirus)	This study reveals that a serotype chimeric infectivity-enhanced CRAd, Ad5/3-24, could be a viable and safe treatment option for recurrent ovarian cancer patients.	[30]
Phase I study of a tropism-modified conditionally replicative adenovirus for recurrent malignant gynecologic disease	Gynecologic malignancy	Ad5/3-(delta)24 (adenovirus)	The feasibility, safety, possible antitumor response, and biological activity of this method in ovarian cancer are demonstrated in this study, which is the first to investigate an infectivity-enhanced CRAd in human cancer. More research on infectivity-enhanced virotherapy for malignant gynaecologic disorders is needed.	[31]
Phase 1 Integrin targeted oncolytic adenoviruses Ad5-D24-RGD and Ad5-RGD-D24-GMCSF for treatment of patients with advanced chemotherapy refractory solid tumors	Advanced solid tumors resistant to chemotherapy	Ad5-D24-RGD and Ad5-RGD-D24-GMCSF (adenovirus)	In a radiological study, all patients treated with Ad5-D24-RGD showed disease progression, albeit 3/6 experienced transient reductions or stabilization of marker levels. ELISPOT was used to demonstrate induction of tumor and adenovirus specific immunity in Ad5-RGD-D24-GMCSF-treated patients. RGD-modified oncolytic adenoviruses with or without GMCSF appear to be safe for clinical testing.	[32]
Phase 1 Antiviral and antitumor T cell immunity in patients treated with GM-CSF-coding oncolytic adenovirus	Advanced solid tumors	CGTG-102 (Ad5/3-delta24-GMCSF (adenovirus)	There findings are the first to relate antiviral immunity to antitumor immunity in humans, suggesting that oncolytic viruses could play a key role in cancer immunotherapy.	[33]
Immunological data from cancer patients treated with Ad5/3-E2F-∆24-GMCSF suggests utility for tumor immunotherapy	Advanced solid tumors	CGTG-602 (Ad5/3-E2F-delta24-GMCSF) (adenovirus)	Tumor biopsies revealed that after therapy, immune cells, particularly T-cells, accumulated in tumors. Tumor RNA expression analysis revealed immune activation and metabolic alterations as a result of virus replication.	[34]
Phase I trial of CV706, a replication-competent, PSA selective oncolytic adenovirus, for the treatment of locally recurrent prostate cancer	Prostate cancer	CV706 (PSA selective adenovirus)	Taken together, the findings show that CV706 may be safely delivered intraprostatically to patients, even at high dosages, and the data also suggest that CV706 has enough clinical efficacy, as measured by changes in blood PSA, to support further clinical and laboratory research.	[35]
Phase I trial of intravenous CG7870, a replication-selective, prostate-specific antigen-targeted oncolytic adenovirus	Hormone refractory metastatic prostate cancer	CG7870 (adenovirus)	There were no partial or complete PSA responses detected; however, 5 patients showed a 25 percent to 49 percent drop in serum PSA after a single therapy, including 3 of 8 patients at the highest dose levels.	[36]
First-in-human phase 1 study of CG0070, a GM-CSF expressing oncolytic adenovirus, for the treatment of non-muscle-invasive bladder cancer	Bladder cancer (non-muscle)	CG0070 (GM-CSF expressing adenovirus)	Intravesical CG0070 showed a tolerable safety profile as well as anti-bladder cancer activity. The expression of the granulocyte-monocyte colony-stimulating factor transgene and CG0070 replication have also been suggested.	[37]
Phase I study of KH901, a conditionally replicating granulocyte-macrophage colony-stimulating factor: armed oncolytic adenovirus for the treatment of head and neck cancers	Advanced solid tumors	KH901 (GM-GSF Expressing Adenovirus)	These preliminary findings demonstrated that intratumoral injection of KH901 was possible, well tolerated, and related to biological activity, indicating that more research into KH901, particularly in combination with systemic chemotherapy, is needed.	[38]
Oncolytic adenovirus ICOVIR-7 in patients with advanced and refractory solid tumors	Advanced solid tumors	ICOVIR-7 (adenovirus)	In total, 9 of the 17 evaluable patients showed objective evidence of anticancer efficacy. In radiological analysis, 5 of the 12 evaluable patients had tumor size stabilization or reduction. One partial response, two modest responses, and two cases of stable disease were observed in patients who had been experiencing increasing disease prior to treatment. In conclusion, ICOVIR-7 therapy appears to be safe and has anticancer action, making it a suitable candidate for additional clinical trials.	[39]
Phase I open-label, dose-escalation, multi-institutional trial of injection with an E1B-Attenuated adenovirus, ONYX-015, into the peritumoral region of recurrent malignant gliomas, in the adjuvant setting	Malignant glioma	ONYX-015 (adenovirus)	The median time to death was 6.2 months (range: 1.3 to 28.0+ months). One patient has shown regression of interval-increased enhancement, whereas the other has not progressed. After more than 19 months of follow-up, 1/6 of 109 pfu recipients and 2/6 of 1010 pfu recipients are still alive. A lymphocytic and plasmacytoid cell infiltration was found in two individuals who had a second resection three months after receiving ONYX-015 injection. At doses up to 1010 pfu, ONYX-015 injection into glioma cavities is well tolerated.	[40]
Phase I trial of intravenous infusion of ONYX-015 and Enbrel in solid tumor patient	Advanced cancers	ONYX-015 (adenovirus)	In the absence of enbrel, the area under the curve measurements show a significantly higher amount of TNF-induction and faster clearance at cycle 2. It is suggested that more research is conducted.	[41]
Phase I study of Onyx-015, an E1B attenuated adenovirus, administered intratumorally to patients with recurrent head and neck cancer	Recurrent head and neck cancer	ONYX-015 (adenovirus)	Despite being below detectable levels at 24 h, viral DNA was detected in plasma or sputum of four patients on days 7 and 14 after therapy, implying viral replication. The injected malignant lesion in one patient only responded somewhat. At day 56 after treatment, seven patients had stable disease, as defined by the Response Evaluation Criteria in Solid Tumors (RECIST). The telomelysin was tolerated well. It was suggested that there was evidence of anticancer action.	[42]
Phase I study of telomerase-specific replication competent oncolytic adenovirus (telomelysin) for various solid tumors	Advanced solid tumors	H103 (Adenovirus expressing HSP70)	Despite being below detectable levels at 24 h, viral DNA was detected in plasma or sputum of four patients on days 7 and 14 after therapy, indicating viral replication. The injected malignant lesion in one patient responded partially. At day 56 after treatment, seven patients met the criterion of stable disease as defined by the Response Evaluation Criteria in Solid Tumors (RECIST). Telomelysin was tolerated well. There was some evidence of anticancer action.	[43]
Phase I study of replication-competent adenovirus-mediated double suicide gene therapy for the treatment of locally recurrent prostate cancer	Prostate	Ad5-CD/TKrep (adenovirus)	The findings reveal that intraprostatic administration of the replication-competent Ad5-CD/TKrep virus followed by 2 weeks of 5-fluorocytosine, and ganciclovir prodrug therapy may be completed safely in people and that biological activity can be shown.	[44]
Phase I trial of replication-competent adenovirus-mediated suicide gene therapy combined with IMRT for prostate cancer	Primary or metastatic liver cancer	Ad5-yCD/mutTKSR39rep-ADP (adenovirus)	The findings show that this exploratory method is safe, and they suggest the possibility that it could improve the outcome of conformal radiation in some patient groups.	[45]
Phase I trial oncolytic measles virus in cutaneous T cell lymphomas mounts antitumor immune responses in vivo and targets interferon-resistant tumor cells	Cutaneous T cell Lymphoma	MV (Measles Virus, Edmonston-Zagreb strain)	Clinical responses obtained from the well-tolerated MV therapy. Immunohistochemistry and reverse transcriptase-polymerase chain reaction (RT-PCR) analysis of biopsies before and 11 days after injection revealed local viral activity with positive staining for MV nucleoprotein (NP), an increase in the interferon (IFN-)/CD4 and IFN-/CD8 mRNA ratios, and a reduced CD4/CD8 ratio. After treatment, all the patients had a higher anti-measles antibody titer. CTCLs are a suitable target for an MV-based oncolytic treatment, according to the findings.	[46]
Phase I trial of systemic administration of Edmonston strain of measles virus genetically engineered to express the sodium iodide symporter in patients with recurrent or refractory multiple myeloma	Relapsed and refractory multiple myeloma	MV-NIS (measles virus with sodium iodide symporter)	Before being eliminated by the immune system, MV-NIS can replicate. Oncolytic viruses are a promising new method for infecting and killing disseminated myeloma cells.	[47]
Phase I trial of intraperitoneal administration of an oncolytic measles virus strain engineered to express carcinoembryonic antigen for recurrent ovarian cancer	Taxol and platinum-refractory recurrent ovarian with normal CEA levels	MV-CEA (Measles virus, Edmonston strain)	They have demonstrated both safety and early, promising biological activity in this first human study of an oncolytic MV strain in the treatment of recurrent ovarian cancer. Further research into this oncolytic virus platform in the treatment of recurrent ovarian cancer is needed.	[48]
Phase I trial of Seneca Valley Virus (NTX-010) in children with relapsed/refractory solid tumors	Pediatric patients with neuroblastoma, rhabdomyosarcoma, rare tumors with NET features	NTX-010 (Seneca Valley Virus)	In metastatic melanoma patients, reovirus treatment was well tolerated, and viral replication was seen in biopsy samples. Preclinical evidence of synergy with taxanes and platinum compounds.	[49]
Phase I clinical study of Seneca Valley Virus (SVV-001), a replication-competent picornavirus, in advanced solid tumors with neuroendocrine feature	Advanced solid tumors with neuroendocrine features	SVV-001 (Seneca Valley Virus, a picornavirus)	In metastatic melanoma patients, reovirus treatment was well tolerated, and viral replication was seen in biopsy samples. Preclinical evidence of synergy with taxanes and platinum compounds.	[50]
Randomized phase IIB evaluation of weekly paclitaxel versus weekly paclitaxel with oncolytic reovirus	Ovarian, tubal, or peritoneal cancer	Reolysin (Reovirus)	In metastatic melanoma patients, reovirus treatment was well tolerated, and viral replication was seen in biopsy samples. Preclinical evidence of synergy with taxanes and platinum compounds.	[51]
Phase II trial of intravenous administration of Reolysin (^®^) (Reovirus Serotype-3-dearing Strain) in patients with metastatic melanoma	Melanoma	Reolysin (Reovirus)	In metastatic melanoma patients, reovirus treatment was well tolerated, and viral replication was seen in biopsy samples. Preclinical evidence of synergy with taxanes and platinum compounds.	[52]
Prospective randomized phase 2 trial of intensity modulated radiation therapy with or without oncolytic adenovirus-mediated cytotoxic gene therapy in intermediate-risk prostate cancer	Prostate	Ad5-yCD/mutTKSR39rep-ADP adenovirus	In males with intermediate-risk prostate cancer, combining OAMCGT with IMRT does not increase the most prevalent adverse effects of prostate radiation therapy and predicts a clinically relevant reduction in positive biopsy results at 2 years.	[53]
Intraprostatic distribution and long-term follow-up after AdV-tk immunotherapy as neoadjuvant to surgery in patients with prostate cancer	Prostate	AdV-tk (also known as a aglatimagene besadenovec or ProstAtak	In vivo transrectal ultrasonography guided instillation of an adenoviral vector into four sites in the prostate was a simple outpatient operation that was well tolerated and resulted in widespread dispersion throughout the intraprostatic tumor mass. There was no major acute or late toxicity associated with AdV-tk. The likelihood of a prolonged immune response to residual disease was suggested by PSA and disease progression trends.	[54]
Phase II multicentre study of gene-mediated cytotoxic immunotherapy as adjuvant to surgical resection for newly diagnosed malignant glioma	Glioma	AdV-tk (aka aglatimagene besadenovec or ProstAtak	In newly diagnosed malignant gliomas, safe integration of gene-mediated cytotoxic immunotherapy (GMCI) with standard care improves survival, especially in patients with less residual disease, fostering further research and GMCI testing.	[55]
A controlled trial of intertumoral ONYX-015, a selectively replicating adenovirus, in combination with cisplatin and 5-fluorouracil in patients with recurrent head and neck cancer	Recurrent squamous cell head and neck cancer	ONYX-15 (adenovirus)	High response and complete recovery rates were observed in treated tumors, with no progression after six months and tolerable side effects, alongside evidence of tumor-specific viral multiplication and necrosis post-therapy.	[56]
Phase I Trial of an ICAM-1-Targeted Immunotherapeutic-Coxsackievirus A21 (CVA21) as an Oncolytic Agent Against Non-Muscle-Invasive Bladder Cancer	Non-muscle invasive bladder cancer	Coxsackievirus A21 (CVA21)	CAVATAK’s efficacy, safety, and unique immunological impact position it as an innovative treatment for non-muscle-invasive bladder cancer (NMIBC).	[57]

**Table 2 cancers-15-04304-t002:** DNA content (%) in different cell cycle stages (Control-C, MMR Vaccine Treated-T).

Cell Type	G0/G1	S	G2/M
BV2-C	53.3%	12.7%	23.7%
BV2-T	68.6%	11.4%	5.7%
U87MG-C	57.3%	8.7%	21.2%
U87MG-T	61%	8.0%	19.3%
U138G-C	57.5%	15.0%	26.5%
U138G-T	63%	14.0%	23.2%
GBM10-C	66.45%	26.3%	5.3%
GBM10-T	62.9%	27.8%	7.3%
GBM23-C	77.9%	14.9%	5.6%
GBM23-T	81.7%	12.4%	4.5%

**Table 3 cancers-15-04304-t003:** Common genes in early embryonic development and certain cancer types during the adult stage.

Name of Gene	Role in Embryonic Development	Expression in Cancers	References
TWIST1	This gene is involved in cranial suture closure during skull development and regulates neural tube closure, limb development, and brown fat metabolism	TWIST1 enhances GBM invasion in concert with mesenchymal change not involving the canonical cadherin switch of carcinoma EMT., Breast, bladder, pancreatic, prostatic, gastric, etc.	[72,73,74]
Trism28	Oocyte and early embryo	Glioma, Breast, Liver, prostatic, and gastric cancers	[75,76]
Nodal	Early inner cell mass	Glioblastoma, Breast, melanoma, and prostatic cancer	[77,78]
Cripto-1	Gastrulation stage, nascent primitive streak, and mesoderm	Glioblastoma multiforme, breast, colon, and lungs cancer	[79,80]
ROR1	Head mesenchyme	Glioma, Leukaemia, Lymphoma, multiple myeloma, and breast cancer	[81,82]
Birc5	Distal bronchiolar epithelium of the lungs	Low-Grade Glioma, prostatic, uterine, renal, and hepatocellular carcinoma	[83,84]
Nrf2f	Contribute to numerous somatic cell types in the testis	Breast cancer, abdominal tumor	[85,86]
Tbx2	Coordinate cell fate, patterning, and morphogenesis of a wide range of tissues and organs including limbs, kidneys, lungs, mammary glands, heart, and craniofacial structures	Melanoma, small cell lung carcinoma, breast, pancreatic, liver, and bladder cancers	[87]
Alpha feto protein (AFP)	Collaborating with estrogen, it safeguards the fetus from maternal estrogen in circulation and hinders the breakdown of hormone molecules. Additionally, it plays a role in immunosuppression, shielding the fetus from the maternal immune system. Moreover, it fosters the growth and specialization of the developing fetus.	Plays a vital role in both triggering the growth and advancement of hepatocellular carcinoma. Used as a biomarker for diagnosis of HCC, testicular, and ovarian cancer.	[88,89]
Carcino embryonic antigen (CEA)	CEA works as a cellular adhesion factor in organ development.	Used as a tumor biomarker in liver, colorectal, and gastrointestinal cancer	[90,91]

## Data Availability

The datasets used and/or analyzed during the current study are available from the corresponding author on reasonable request.

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
