# Peer review of "Anticancer Activity of Measles–Mumps–Rubella MMR Vaccine Viruses against Glioblastoma"

_cancers, 2023, doi:10.3390/cancers15174304_

Round 1
Reviewer 1 Report
Referee’s remarks on “Anticancer activity of Measles Mumps Rubella MMR-vaccine 2 viruses against glioblastoma” submitted for publication in the journal Cancers
Virotherapy against various kinds of cancer been extensively investigated and varied degrees of promise reported. Cleveland Clinic advertises the clinical application of MMR vaccine for treatment of glioblastoma.
Recent papers in related area
Viruses. 2023 Feb; 15(2): 547.
Published online 2023 Feb 16. doi: 10.3390/v15020547
PMCID: PMC9958853
PMID: 36851761
Recent Developments in Glioblastoma Therapy: Oncolytic Viruses and Emerging Future Strategies
Azzam Hamad,1,* Gaukhar M. Yusubalieva,2,3 Vladimir P. Baklaushev,2,3,* Peter M. Chumakov,1 and Anastasiya V. Lipatova1
Review
Cancers (Basel)
. 2021 Feb 4;13(4):614.
doi: 10.3390/cancers13040614.
Personalizing Oncolytic Virotherapy for Glioblastoma: In Search of Biomarkers for Response
Eftychia Stavrakaki 1, Clemens M F Dirven 1, Martine L M Lamfers 1
Affiliations expand
- PMID: 33557101
- PMCID: PMC7913874
- DOI: 10.3390/cancers13040614
Free PMC article
· Tumor-Associated Macrophages/Microglia in Glioblastoma Oncolytic Virotherapy: A Double-Edged Sword.
Int J Mol Sci. 2022 Feb 4;23(3):1808. doi: 10.3390/ijms23031808.PMID: 35163730 Free PMC article. Review.
· Oncolytic Zika Virus: New Option for Glioblastoma Treatment.
DNA Cell Biol. 2023 Jun;42(6):267-273. doi: 10.1089/dna.2022.0375. Epub 2022 Nov 9.PMID: 36350682 Review.
· Cancer stem cell plasticity in glioblastoma multiforme: a perspective on future directions in oncolytic virotherapy.
Future Oncol. 2020 Oct;16(28):2251-2264. doi: 10.2217/fon-2019-0606. Epub 2020 Aug 2.PMID: 32744059 Review.
· Intraarterial delivery of virotherapy for glioblastoma.
Neurosurg Focus. 2021 Feb;50(2):E7. doi: 10.3171/2020.11.FOCUS20845.PMID: 33524944 Review.
· Synthetic and systems biology principles in the design of programmable oncolytic virus immunotherapies for glioblastoma.
Neurosurg Focus. 2021 Feb;50(2):E10. doi: 10.3171/2020.12.FOCUS20855.PMID: 33524942 Free PMC article. Review.
Some suggested corrections
|
line |
Present form |
Suggested |
|
Abstract |
|
|
|
13. |
….followings objectives;… |
….following objectives: (use a colon)… |
|
13. |
..to determine the role of cancer virotherapy.. |
..to evaluate the therapeutic effect of measles virus vaccine strains on cancer..(use commas instead of semi-colons between the objectives) |
|
14. |
…on the function of… |
…on the application of.. |
|
15-16 |
..Our findings……(GBM) |
Our findings highlight the therapeutic potential of MMR vaccine strain for the treatment of glioblastoma (GBM). |
|
29-30 |
To the best of our knowledge, insofar no study reported anticancer activity of MMR vaccine against glioblastoma. |
Therapeutic potential of MMR vaccine has been found promising in earlier studies as well. |
|
34. |
Oncolytic virotherapy for cancer treatment has lately been discovered. |
Delete |
|
40-41 |
…..dengue fever [2]. |
…..dengue fever viruses [2]. |
|
46 |
……takeovers.. |
…takes over.. |
|
51. |
…paramyxovirus, measles,…. |
…measles virus.. (measles virus is a paramyxovirus) |
|
Table 1 |
References may be presented in the last column instead of first. First column may be serial number |
|
|
Table 1 |
Name of Trail |
Phase of trial |
|
Figure 2 |
Size bar be inserted in the figures |
|
|
199 |
… to divide inside the cells. |
… to replicate inside the cells. |
|
228. |
..than that of detected.. |
…than that detected… |
|
258-284 |
This section cannot be included under results for the present study and may be moved to discussion. |
|
|
|
|
|
Virotherapy against various kinds of cancer has been extensively investigated and varied degrees of promise have been reported. Cleveland Clinic advertises the clinical application of the MMR vaccine for treating glioblastoma.

I have listed some of the typographical/compositional errors. The manuscript may need further with copy line editing.
Author Response
Dear Editors,
We would like to thank you for considering the manuscript entitled:
“Anticancer activity of Measles Mumps Rubella MMR-vaccine viruses
against glioblastoma” by Z. Khalid et al., and for sharing the Reviewers’ comments that certainly helped in improving the quality of the manuscript (Cancers 2556793). We appreciated the Reviewers’ comments, and we revised the manuscript accordingly. Please find enclosed to the submission of the revised version of the manuscript the point-by point reply to the Reviewers’ comments. For clarity’s sake, changes in the revised MS are wrote in red colour. We hope that the revised version of our MS will be now suitable for publication in Cancers.
Accordingly, we prepared a revised version of the manuscript acknowledging Referees’ and Editor’s comments as below specified:
Reviewer 1:
COMMENT 1. Virotherapy against various kinds of cancer been extensively investigated and varied degrees of promise reported. Cleveland Clinic advertises the clinical application of MMR vaccine for treatment of glioblastoma.
ANSWER 1. We wish to thank the Reviewer for the kind and positive comments. We have revised the manuscript, taking into account the Reviewer’s concerns below.
COMMENT 2. Recent papers in related area.
ANSWER 2. We thank the Reviewer for the note. We have added the references suggested in the introduction.
COMMENT 3. Some suggested corrections in Abstract, Table 1, Figure 2 and Discussion.
ANSWER 3. We thank the Reviewer for the suggested corrections. We have made the corrections requested in the text.

Reviewer 2 Report
Oncotherapy attracts the attention of many scientists who are looking for new more effective protocols. Viral vaccines have shown their reliability during the coronovirus pandemic and confirmed the possibility of their use in the treatment of oncopathologies. Therefore, the study of new designs of viral vaccines for oncotherapy is extremely important.
The methods are described in detail, the results and discussions are logically constructed, the conclusions are adequate. The manuscript is well illustrated, which makes it easier to understand the material.
The manuscript will be of interest to many readers, including young scientists who are just starting their path in science. It may be accepted for publication after correction of minor typographical errors.
Remarks
Line 183
blue columns) or treated with MMR vaccine for 72 hours (grey columns). (a) Cell viability of normal
Have to be
blue columns) or treated with MMR vaccine for 72 hours (orange columns). (a) Cell viability of normal
Author Response
Reviewer 2:
COMMENT 1. The methods are described in detail, the results and discussions are logically constructed, the conclusions are adequate. The manuscript is well illustrated, which makes it easier to understand the material. The manuscript will be of interest to many readers, including young scientists who are just starting their path in science. It may be accepted for publication after correction of minor typographical errors. Line 183 blue columns) or treated with MMR vaccine for 72 hours (grey columns). (a) Cell viability of normal Have to be: blue columns) or treated with MMR vaccine for 72 hours (orange columns). (a) Cell viability of normal.
ANSWER 1. We wish to thank the Reviewer for the kind and positive comments. We have revised the manuscript, taking into account the Reviewer’s suggestions.
